# OpenReview forum: "Automatic Hallucination Assessment for Aligned Large Language Models via Transferable Adversarial Attacks"
_ICLR.cc/2024/Conference — ICLR 2024 Conference Withdrawn Submission_

### Official Review · Reviewer_ZWCj · 2023-10-31

**Soundness:** 3 good
**Presentation:** 3 good
**Contribution:** 2 fair
**Rating:** 3
**Confidence:** 4

**Summary:**

In this paper the authors present an automatic method to generate new samples that could trigger hallucinations in LMs by perturbing old samples that produced faithful generations. They use two methods for generating new samples - 1) they use a LM to generate an incorrect answer and replace the original answer with it in the context and answer, 2) they augment the context of a sample by adding additional sentences,

**Strengths:**

The authors empirically evaluate 5 different LMs and demonstrate that the performance does drop when the samples are modified with AutoDebug.

**Weaknesses:**

Here are some weaknesses:
* There is limited novelty in this paper. Both changing the answer and adding additional sentences to the context have been previously explored for different NLP use cases. Since, the paper only presents results on one dataset, the empirical exploration is also limited.
* There are many proposed methods for reducing hallucination. The paper does not present results on using any of these methods and evaluating whether the language model still hallucinates.
* No qualitative examples have been provided for category 2 data.

**Questions:**

Here are some questions:
* How much does length increase for category 2 (LM-enriched evidence) data?
* Do other datasets lead to similar results?
* What is the size of the dataset for the results in table 1 and table 2? Section 3 mentions that there are 7189 samples after filtering, but how many of these are identified as seed test cases?
* Have you considered other types of perturbing data like adding arbitrary text or paraphrasing the questions/context?

**Details Of Ethics Concerns:**

No ethics review required.

---

### Official Review · Reviewer_akN9 · 2023-10-31

**Soundness:** 3 good
**Presentation:** 3 good
**Contribution:** 2 fair
**Rating:** 3
**Confidence:** 3

**Summary:**

The paper introduces AutoDebug, a framework using prompt chaining to generate adversarial question-answering examples (based on NaturalQuestions) for evaluating LLMs. The authors find that LLMs are likely that an easy way of eliciting hallucinations is to exploit conflicts between knowledge given in the prompt and their parametric knowledge or (2) cases when the knowledge expressed in the prompt is complex. They also find adversarial QA pairs to be transferable across model families.

**Strengths:**

1. The paper is written clearly and easy to follow
2. I like the framing of hallucinations are adversarial vulnerabilities to certain prompts and the focus on transferrable and human-readable attacks

**Weaknesses:**

1. From what I understand, this is a method paper (introducing a new method, AutoDebug), not an analysis paper. Therefore, I would expect the method to be compared with baseline approaches. I don't see that comparison and it's not clear how notable are the numbers the authors report.  The the method is quote simple I'm not convinced it's a contribution an ICLR-scale contribution.
2. I think the the breadth and depth of experiments is relatively weak the proposed pipeline is pretty narrow in scope: it focuses just on hallucinations, just on QA hallucinations, and on a single QA dataset, and with just a couple of models evaluated. There are no scaling experiments, i.e. changing model size for the same model family. Therefore, it's not clear how scalable is AutoDebug wrt model size.
3. I'm not convinced by the remark that "GPT-4 is more reliable than Alpaca-7B, which suggests that recent efforts in aligning LLMs is promising for developing more trustworthy models". It might very well be the case that GPT-4 is superior due to superior capabilities (e.g. model size), not superior alignment.

**Questions:**

Minor:
Contrasting GPT-4 with ChatGPT is misleading. Both GPT-3.5 and GPT-4 are models used in the ChatGPT product; it's better to refer to these models as GPT-3.5 and GPT-4.

---

### Official Review · Reviewer_6oNo · 2023-10-31

**Soundness:** 3 good
**Presentation:** 3 good
**Contribution:** 2 fair
**Rating:** 5
**Confidence:** 3

**Summary:**

This paper designs two types of probing data based on an existing QA dataset, which are deliberately (“adversarially”) perturbed to trigger hallucinations in LLMs. It then measures how robust common LLMs are against such “adversarial” data. The model’s performance on the proposed data can potentially indicate their reliability (i.e., how likely to hallucinate).

**Strengths:**

1. The problem studied in this work is important. The idea of designing probing data to assess hallucination in LLMs is exciting and practical.
2. Experiments cover a good range of LLMs. The authors also conducted human evaluations to verify the quality of their generated evaluation data.

**Weaknesses:**

1. The design of Category 1 evaluation is unclear and conflicted to me: Does it expect a good model to change the answer accordingly? In other words, does a higher number in Table 2 mean a model is better at updating its answer on the updated evidence accordingly?

    1-a. If that is the case, I don’t think it would be appropriate to call it a hallucination if the model still gives the correct answer regardless of the open-book context. For example, if the question is a common sense question and the modified context is not very relevant, or the wrong answer is far way from correct, I don’t expect the model to change its answer according to the context.

    1-b. On the other hand, if a larger number in Table 2 means the model is less likely to update its answer according to the context, it also doesn’t make sense in some scenarios. For example, I would expect a good model to give an answer corresponding to the context. If the context deliberately presents a wrong answer and models give the corresponding wrong answer, I don’t think this fits the definition of hallucination, because the model’s answer is based on the supporting evidence.
2. The results presentation can be improved to provide more information and better support the claim. For example, in Section 3.2, it says, “The model resistance towards our attack is mostly correlated with its model size”. However, without showing each model’s original (clean) performance and the performance delta caused by the attack, it’s unclear how “robust” a model is or its resistance towards the attack. I’d also like to see separate performance delta on self-attack and cross-attack, which helps demonstrate how transferable the attacks are.
3. I’m unsure if ``AutoDebug” is an accurate name for the proposed method. I would expect a “debugging” method to effectively identify, locate, and address certain issues, while the proposed evaluation mainly seems to trigger/identify the issues.

**Questions:**

1. It’s unclear to me why and how would Category 1 data affect the closed-book performance, because closed-book QA doesn’t rely on the context.
2. As stated in weakness 1, I’m a bit confused about the expected behavior on category 1 data. I would appreciate a clarification on this.

---

### Official Review · Reviewer_mBjG · 2023-11-05

**Soundness:** 3 good
**Presentation:** 3 good
**Contribution:** 2 fair
**Rating:** 5
**Confidence:** 3

**Summary:**

In many scenarios, users expect LLMs to answer questions based on a given source, such as a document. To investigate the extent to which LLMs can be faithful to the given source (i.e., not hallucinate), this paper proposes a method to construct more challenging datasets for more efficient (red team) testing. The method works by first identifying seed questions from existing QA data that the model can answer correctly with the provided source. Then, it modifies these questions by changing parts of the source that answer the question or adding more contextual information to make them more difficult for LLMs. The identification and modification are carried out by prompting a baseline LLM, such as ChatGPT. The authors used this method to generate two variants of the Natural Questions dataset and evaluated the performance of five different LLMs on them. The results show that existing models achieve an accuracy of up to 82% on answer-swapping questions and up to 67% on context-enriching questions, both below human levels.

**Strengths:**

1. This paper addresses a practical and important problem: whether LLMs can faithfully ground their answer on the provided source. The paper provides an efficient approach to stress test LLM's faithfulness on the given sources.
2. The proposed method is interesting. The authors use several techniques to make evaluation more reliable: they consider three metrics in calculating accuracy and employ a human evaluation to ensure the validity of the generated datasets.
3. The paper is generally well-written and easy to read.

**Weaknesses:**

1. This paper's technical contribution is not significant.
    - The paper uses a heuristic method to modify existing datasets, yet it lacks empirical validation and comparative analysis to demonstrate the effectiveness of this method. For example, is the current seed question identification optimal? Would it be more efficient to select questions that the model still answers incorrectly even when open-book? I suggest that the authors add some baselines for comparison, such as the performance of different models on the original dataset, to support their method design.
    - The assessment is only conducted on one QA dataset. How well the method can be extended to other datasets or tasks remains unknown.
    - While the authors use three metrics to calculate accuracy, their consistency with human evaluation is still unclear. It would be better if the authors sample a subset of data to measure the consistency of these three metrics with human evaluation.
2. The results do not provide much insightful implications. Due to the lack of baselines, the reader can only draw unsurprising conclusions, such as GPT-4 being more likely to rely on provided sources to answer questions than Alpaca-7B. Furthermore, the two implications proposed by the authors in the abstract seem more like the natural result of the specific dataset construction. Therefore, I again suggest that the authors design more meaningful baseline data to derive deeper insights.
3. Given the current zero-shot open-ended question prompting, the problem setup of "answer-swapping" might be ambiguous for LLMs. For example, the authors do not make it clear that the model should rely merely on the provided source to answer the question, even if the source is factually incorrect. In this case, if the model's answer is factually correct but contradicts the provided source, why call it a hallucination? I recommend designing prompts more cautiously to define the problem setup clearly.
4. Although failing to answer based on the provided source is indeed an important type of hallucination, there are other types of hallucination, such as being inconsistent with previous answers and providing factually incorrect answers. This paper's method seems only applicable to the first type. This limitation also makes the paper's title, "Automatic Illusion Assessment," somewhat overstated. If possible, the authors may provide more specific examples of extending their methods to other types of hallucination.
5. The font in Figure 2 is too small.

**Questions:**

Please see the Weakness section.